# Plasma Metabolome and Lipidome Associations with Type 2 Diabetes and Diabetic Nephropathy

**DOI:** 10.3390/metabo11040228

**Published:** 2021-04-08

**Authors:** Yan Ming Tan, Yan Gao, Guoshou Teo, Hiromi W.L. Koh, E Shyong Tai, Chin Meng Khoo, Kwok Pui Choi, Lei Zhou, Hyungwon Choi

**Affiliations:** 1Department of Statistics and Applied Probability, Faculty of Science, National University of Singapore, Singapore 117546, Singapore; yanmingtan5@gmail.com (Y.M.T.); choikp@nus.edu.sg (K.P.C.); 2Singapore Eye Research Institute, The Academia, 20 College Road, Singapore 169856, Singapore; gaoyan3540@outlook.com; 3Department of Medicine, Yong Loo Lin School of Medicine, National University of Singapore, Singapore 119228, Singapore; guoshou@nus.edu.sg (G.T.); mdckwlh@nus.edu.sg (H.W.L.K.); mdctes@nus.edu.sg (E.S.T.); mdckcm@nus.edu.sg (C.M.K.); 4Department of Ophthalmology, Yong Loo Lin School of Medicine, National University of Singapore, Singapore 119228, Singapore; 5Ophthalmology and Visual Sciences Academic Clinical Research Program, Duke-NUS Medical School, National University of Singapore, Singapore 169857, Singapore

**Keywords:** diabetic nephropathy, data independent acquisition, prostaglandins, phospholipids, uremic toxins, oxidative stress

## Abstract

We conducted untargeted metabolomics analysis of plasma samples from a cross-sectional case–control study with 30 healthy controls, 30 patients with diabetes mellitus and normal renal function (DM-N), and 30 early diabetic nephropathy (DKD) patients using liquid chromatography-mass spectrometry (LC-MS). We employed two different modes of MS acquisition on a high-resolution MS instrument for identification and semi-quantification, and analyzed data using an advanced multivariate method for prioritizing differentially abundant metabolites. We obtained semi-quantification data for 1088 unique compounds (~55% lipids), excluding compounds that may be either exogenous compounds or treated as medication. Supervised classification analysis over a confounding-free partial correlation network shows that prostaglandins, phospholipids, nucleotides, sugars, and glycans are elevated in the DM-N and DKD patients, whereas glutamine, phenylacetylglutamine, 3-indoxyl sulfate, acetylphenylalanine, xanthine, dimethyluric acid, and asymmetric dimethylarginine are increased in DKD compared to DM-N. The data recapitulate the well-established plasma metabolome changes associated with DM-N and suggest uremic solutes and oxidative stress markers as the compounds indicating early renal function decline in DM patients.

## 1. Introduction

Diabetic nephropathy (DKD) is a multifactorial microvascular complication of diabetes mellitus (DM) [1]. Declining kidney function of DM patients may manifest itself with or without notable proteinuria, leading to chronic kidney disease and end-stage renal disease in some patients over an average time span of several years [2,3]. In the clinical practice around the world, formal diagnosis of DKD almost exclusively relies on invasive renal biopsy, and only a limited proportion of DM patients with renal impairment undergoes biopsy. For this reason, there has been a long-standing interest in introducing circulating biomarkers as a diagnostic criterion or a monitoring tool for clinical management of the diseases beyond the current gold standard, i.e., estimated glomerular filtration rate (eGFR) and its past trajectories calculated from serum creatinine or cystatin-C with adjustments for age, gender, and race, and urine albumin/creatinine ratio (uACR). As discussed in recent extensive reviews of diagnostic biomarkers of DKD [4] and clinical management [5], a growing body of evidence from prospective and cross-sectional studies supports the utility of plasma or urinary protein markers originating from various anatomical sites of origin, such as cystatin C [6,7], copeptin [8], kidney injury molecule-1 (KIM-1) [9,10], neutrophil gelatinase-associated lipocalin (NGAL) [7], TNF receptors, as well as microRNAs.

Metabolomics studies have also contributed to the repertoire of candidate biomarkers for non-diabetic chronic kidney disease (CKD) such as IgA nephropathy, revealing amino acids and their metabolites, tryptophan metabolites, uric acid and other purine metabolites, oxidative stress, and lipids and acylcarnitines as promising markers [11]. However, the presence of DM influences the circulating levels of metabolites and the search for DKD-specific markers may require the direct comparison of healthy non-DM subjects, DM patients with normal renal function, and DKD patients. Moreover, in contrast to the plasma proteomics studies for DKD biomarker discovery, most discovery-phase metabolomics studies were conducted before 2015. This was an era when untargeted mass spectrometry used to be performed without systematic MS/MS fragmentation, with compound identification depending on the precursor ion information and proprietary libraries from instrument vendors. MS technology and bioinformatics software have substantially advanced over the past few years, and it is worth revisiting the global identification of compounds and semi-quantification by untargeted LC-MS in this arena.

To address these gaps, we applied a modern untargeted metabolomics workflow to a cross-sectional case control study of 90 participants. Here we compare metabolite levels among healthy controls (N = 30), DM patients with normal renal function (DM-N, average eGFR 108.6 mL/min/1.73m^2^, N = 30), and DM patients with early nephropathy patients (DKD, average eGFR 72.6 mL/min/1.73m^2^, N = 30). We first employed data-dependent acquisition (DDA) MS to all ninety plasma samples to achieve a high rate of MS/MS-based identification. We subsequently re-analyzed the samples using data-independent acquisition (DIA) MS for consistent semi-quantification across all samples.

In addition, we demonstrate a network-based multivariate data analysis approach as a feature exploration tool for the identification of differential metabolites for DM-N and DKD. This approach is particularly useful for the analysis of omics data sets in which the measurements are highly correlated. At the same time, however, we emphasize that the technical advances presented in this work do not address the fact that the data are generated from a cross-sectional case study, and refrain from interpreting the altered metabolite levels between groups as causal agents in the pathogenesis of DM and DKD.

In what follows, we structured the groups comparisons in two stages. To identify the changes in plasma levels of metabolites and lipids, we first compared the 60 DM patients to the 30 control, and later compared the 30 DKD subjects and the 30 DM-N patients. In parallel, we first undertook traditional differential analysis using univariate hypothesis testing, and repeated the analysis for metabolite panel selection for the two stages using the supervised multivariate analysis. We describe these results in more detail below.

## 2. Results

### 2.1. Description of the Cohort

The patient characteristics of 90 subjects can be found in Table 1. The DM-N group was derived from patients with type 2 DM with eGFR > 90 mL/min/1.73 m^2^. The DKD groups was derived from patients with type 2 DM with eGFR <90 mL/min/1.73 m^2^. Overall, the duration of DM was 9.8 (6.9) years (8.6 years for DM-N vs. 10.8 years for DKD, *p* = 0.23). Compared to the 60 DM patients, the control samples were entirely Chinese and younger overall. The DM-N group contained more females than the others. As expected from the phenotypes, DM-N and DKD groups showed higher body mass index (BMI) than the controls, with elevated systolic blood pressure and substantially higher glycated hemoglobin (HbA1c) levels. Serum creatinine levels were the highest among the DKD groups, which is the major determinant in the estimation of glomerular filtration rates (eGFR). Lastly, the mean eGFR of the DKD patients was 72.6 with standard deviation of 16. This suggests that in the patients classified to the DKD group, the kidney damage is mild and renal function is still well preserved with eGFR levels corresponding to stage 2 kidney disease.

### 2.2. Compound Identification and Semi-Quantification

We next performed LC-MS analysis of plasma samples in information-dependent (IDA) acquisition, a form of DDA analysis, for compound identification using an ultrahigh-performance liquid reverse phase chromatography (UPLC) coupled to a high-resolution time-of-flight mass spectrometer (see Materials and Methods). We analyzed all 90 samples for DDA acquisition to achieve the most depth in identification and create the most comprehensive MS/MS spectral library from these samples. The analysis was performed in positive and negative ionization modes for all samples. MetaboKit software identified 1233 unique compounds and 449 records of possible in-source fragmentation (ISF) in positive ionization mode, whereas it identified 937 unique compounds and 321 ISF events in negative ionization mode. In both cases, we discarded the ISFs from all downstream analysis. We remark that, at this stage of data curation, the compounds include not only endogenous compounds, but also a number of contaminant chemicals such as EDTA as well as drug compounds.

We next analyzed the same samples with sequential window acquisition of all theoretical mass spectra (SWATH-MS) scans of 25 Da isolation windows (1 Da overlap), a mode of DIA, for metabolite semi-quantification by peak area calculation across the samples (see Materials and Methods, Appendix A). Using the MS/MS spectral library generated from the IDA data with each constituent spectrum marked with adduct, precursor *m/z*, and retention time (RT) information annotated by MetaboKit [12,13], we extracted peak area data at both MS1 and MS2 levels, with MS2-based fragment ion peak areas rolled up to each metabolite using the mapDIA software with default parameters [14]. Unfortunately, fragment ion chromatograms were generally of inferior quality with respect to the smoothness of ion chromatograms in these data, possibly due to a fast gradient in the LC system used in the study. Therefore, all quantitative data are based on the peak areas of precursor ions from here onwards. We chose all compounds with missing data up to 10 percent of samples (N = 9), and we imputed the values with the 90% of the minimum observed peak areas in each compound.

This process produced 1088 endogenous compounds with semi-quantification across the 90 samples. These compounds encompass a diverse class of metabolites. Although standard methanol-based extraction is not particularly geared to the identification of lipids, 55% of the compounds with MS/MS matching were lipids, including prostaglandins, phospholipids, ceramides, but few neutral lipids such as mono, di, and triacylglycerols as well as sterols were detected. Other detected compounds included fatty acids (8%), and organic acids, amino acids (including their derivatives), and short peptides all accounting for approximately 5% of the compounds, and acylcarnitines, nucleosides/nucleotides, sugars, glycans, purine metabolism intermediates, and uremic toxins were from other classes of metabolites. The remaining ~12% of the detected metabolome includes trace quantities of alcohols, aldehydes, alkaloids, ketones, and organic bases.

Meanwhile, we were also able to identify 125 compounds that are either exogenously synthesized or therapeutic agents (Appendix A). Exclusion of potential drug or exogenous compounds is crucial for the downstream analysis in this cohort, since they also contribute to the separation of the sample groups. For example, we observed that diabetes medications metformin and sitagliptin were elevated only among the DM patients, as expected. We also observed the antiplatelet therapeutic clopidogrel, the anti-hypertension medications losartan, irbesartan and valsartan, an ACE inhibitor medication enalapril, blood lipid lowering atorvastatin, sulfonylurea-class glipizide and gliclazide among the DM patients, but mostly in DKD patients. However, the most consistently administered medication was metformin, and other compounds were mostly detected above the noise level in fewer than 10% of the subjects. We excluded these compounds from the analysis as they are likely a consequence of the disease or subsequent treatment rather than causal markers of the diseases (Appendix A).

Finally, we explored the correlation structure between the endogenous compounds and the clinical parameters collected from the patients, including age, gender, race, body mass index, systolic blood pressure, HbA1c, eGFR, serum creatinine, total cholesterol, total triglycerides, HDL and LDL by series of univariate linear regression analysis. Appendix A shows multiple testing corrected significance values (q-values) for these pairwise associations. According to this screen, other than the three variables directly associated with the disease pathology (HbA1c, eGFR, and serum creatinine), age was the only risk factor with statistically significant association with 24 compounds, some of which are prioritized below. As such, we will include age as a potential confounder in the multivariate analysis in Section 2.5 below.

### 2.3. Diabetes Mellitus Is Associated with Circulating Levels of Organic Acids and Lipids

We next performed differential abundance analysis using univariate hypothesis testing. As mentioned earlier, we first compared 60 DM patients and 30 controls (Figure 1, Appendix A). At 5% false discovery rate and fold change of magnitude greater than 25%, 108 compounds were differentially abundant. Among these, a total of 48 compounds could be classified as lipids. A multitude of eicosanoids including prostaglandins and leukotriene B4, and lysophosphatidylethanolamines (LPEs) were significantly higher in the DM patients, while ether-linked phosphatidylethanolamines (PEs) were lower. A number of amino acid derivatives, including fructosyl isoleucine, fructosyl phenylalanine, PAG, acetylphenylalanine, methyllysine, asymmetric dimethylarginine (ADMA), 3-phenylpropionylglycine, N-palmitoylglycine and manoyltryptophan, as well as numerous dipeptides were elevated in the DM patients. In terms of carbohydrates, we observed elevation of sucrose and galactouronic acid, an oxidized form of D-galactose in the diabetes groups, but reduction in 2-deoxy-D-glucose, an inhibitor of glucose-6-phosphate production through glycolysis. Last but not least, the concentrations of adenine nucleotides (AMP, ADP), a purine nucleotide (deoxyguanosine triphosphate, or DGTP), and a nucleoside (5-methyluridine) were higher in the DM groups.

While most of the aforementioned compounds are consistently elevated in the two DM groups compared to the controls, few are directly involved in the pathophysiology of diabetic nephropathy to the best of our knowledge. By contrast, we observed a few uremic solutes that were already elevated in the DM-N patients that stay elevated in the DKD patients, including tryptophan catabolites indolelactic acid and indole-3-propionic acid, a theophylline metabolite 1,3-dimethylurate, and urea. Interestingly, the neuroprotective antioxidant 3-indolepropionic acid (also known as 1H-indol-3-propanoic acid), produced by human microbiota in the gastrointestinal tract, was lower in the DM-N and DKD groups than in the control group. Lastly, catabolic products of vitamin B6 (pyridoxine), namely 4-pyridoxic acid and pyridoxal [15], were significantly elevated in the DM-N group, indicating that some of the DM-N patients already showed metabolite profiles reflecting renal insufficiency.

### 2.4. DKD Is Associated with Plasma Uremic Toxins

We next examined the difference in relative abundance between early DKD patients and DM-N patients. In this analysis, the total number of significant findings was relatively small. As such, we allowed a lenient significance threshold controlling the FDR at 0.2, keeping the 25% minimum fold change requirement (Appendix A). This resulted in 17 significant compounds (Figure 2), and all these metabolites were elevated in the DKD group compared to the DM-N group, with some compounds showing increasing concentrations from controls to DM-N and to DKD.

The most statistically significant abundance differences included 3-methylhistidine, a marker of muscle protein catabolism in DM-N; glutamine and PAG, a feature of urea cycle disorders; 3α-etiocholanediol (5β-androstane-3α,17β-diol), epiandrosterone (5α-androsten-3β-ol-17-one), 16-dehydropregnenolone (5,16-pregnadien-3β-ol-20-one), metabolites of androgen and testosterone; indoxyl sulfate and hippurate, both uremic toxins; xanthine, a purine degradation product; and an ether-linked phosphatidylcholine 34:3e, all of which had increased average concentrations from control to DM-N and to DKD (FDR < 0.1). Additional compounds in the next tier of statistical significance (FDR < 0.2) included ADMA, a well-known nitric oxide synthase inhibitor, acetylphenylalanine, an amphipathic metabolite of phenylalanine, and 1,3-dimethylurate, a theophylline metabolite.

### 2.5. Partial Correlation Network Analysis for further Prioritization of Candidate Markers

A drawback of statistical differential abundance analysis is that, despite assuring fold changes between groups, not all compounds are directly correlated with essential clinical parameters such as HbA1c and eGFR within each comparison group. To further prioritize the compounds with correlation with these parameters, we have performed a supervised network-based differential analysis. To this end, we first estimated a partial correlation network of 1088 endogenous compounds and all clinical parameters excluding gender and race from the entire data using sparse Gaussian graphical model called graphical lasso [16], which resulted in 23,456 and 20,731 network edges of non-zero partial correlations, a form of confounding-free correlation, when analyzing all 90 subjects and the 60 subjects with DM, respectively. Using the non-zero partial correlations as a pseudo-network, we used the iOmicsPASS algorithm [17] to identify subnetworks associated with DM, and subsequently with early DKD (Appendix A).

Figure 3A shows that several major hub metabolites underlie the confounding-free correlation network in the comparison between 60 DM patients and 30 controls, including HbA1c measured by an immunoassay, and metabolites AMP, DGTP, sucrose, and ADMA measured by the SWATH-MS analysis. With the exception of 2-deoxy-D-glucose, the plasma levels of these compounds were higher in the subjects with DM. Among these, AMP, ADMA, and sucrose exhibited nonzero partial correlations with HbA1c, the main diagnostic criterion for type 2 diabetes, as verified by the heatmap of relative abundance values of individual compounds (Figure 3B). Recalling that the control subjects were younger than the DM patients by 16 years, on average, a caveat in this interpretation is that age has non-trivially large partial correlations with AMP and AMDA, implying that the elevated levels of AMP is in part due to the older age of 60 subjects with DM. However, the selected edges in the network represent the residual associations that contribute to the discrimination of sample groups. Accordingly, a part of the joint variation in the aforementioned compounds still represents age-independent effects of DM.

Likewise, Figure 4A demonstrates that the eGFR, the main parameter associated with albuminuria, has the strongest partial correlations with several metabolites already verified by the univariate hypothesis testing above, including glutamine, (microbiome-associated) PAG, indoxyl sulfate, hippurate, and 3-methylhistidine. However, the network-based multivariate analysis prioritized several other compounds such as a tryptophan, metabolite kynurenine, hippuric acid and 4-hydroxyhippuric acid as additional candidates associated with the discriminant factors for DKD from DM-N. These markers were barely missed by the statistical significance threshold in the univariate analysis, but were restored based on their stronger independent correlation to the key clinical parameter eGFR (Figure 4B).

## 3. Discussion

In this work, we used DDA (IDA) acquisition to create a customized MS/MS spectral library, and subsequently performed DIA (SWATH-MS) analysis for reproducible re-identification and semi-quantification across a large number of samples. This workflow of combining both DDA and DIA from the same samples was seamlessly carried out by a data processing software MetaboKit [13], with the help of reference MS/MS spectral libraries that constantly evolve and are made publicly available to the metabolomics and lipidomics research community by various parties. However, we note that it is not always necessary to perform DDA on all samples. In routine practice, samples pooled from a selected subset of samples often provide enough information to build a spectral library with DDA. Thereafter, as long as the same type of LC system is used, the MetaboKit software assists the user to build a maturing customized spectral library, with which future analysis can be performed using DIA only for semi-quantification.

We also showcased a network-based multivariate analysis framework via iOmicsPASS, rather than the conventional hypothesis testing-based filtering and interpretation of data or multivariable regression modeling approaches that often fail to deal with the multicollinearity problem in correlated high-dimensional data sets. However, it is important to note that we used the method as a feature exploration tool to capture significantly discriminative, correlated variations from a systems biology point of view, and the reported network signatures of DM and DKD, reported in Figure 3 and Figure 4, respectively, were not validated in independent cohorts. It is thus possible that a good proportion of weak edges, i.e. those with partial correlations close to zero, may not be reproduced if the partial correlation network analysis were to be repeated in independent data sets, and the nodes connected by thin edges (weak co-expression scores) may not be selected in future data sets. However, we expect the major hubs in the networks, such as AMP and ADMA, to be highly discriminative features of the network signatures in other studies.

A number of plasma and urine metabolomics studies in DKD and non-diabetic CKD have previously been conducted using GC-MS or LC-MS techniques [18,19,20]. Most metabolites we reported to be associated with DKD, or closely related metabolites, have already been captured with concordant directions of change by respective studies, including compounds involved in urea and ammonia metabolism and excretion [21], gut microbiome-associated tryptophan metabolism [22], uremic toxins [23,24,25,26], glutamate and PAG excretion in urine [27], and uremic solutes also produced by the gut microbiome [24,28]. On the other hand, our metabolome coverage had a limitation of its own. For example, our experiment did not capture some of the well-known free fatty acids and organic acids and intermediates from renal organic ion transport and mitochondrial activity such as TCA cycle, as previously reported [11,20]. In certain compounds, the direction of changes we observed in our experiments (e.g., glucuronide, hippuric acid) was inconsistent with those reported in a previous urine metabolomics study [29], although it is possible that the changes in blood and urine may not always coincide.

Several interesting observations arose from our data. First, the circulating level of AMP is elevated in subjects with DM with a high correlation with HbA1c. The increase in the plasma levels of adenine nucleotides insinuates subsequent overactivation of the fuel-sensing enzyme AMP-activated protein kinase (AMPK) in insulin resistant individuals [30,31], or AMP elevation may predate the hyperglycemia as an upstream regulator of glucose uptake in the whole body [32,33]. Therefore, assays of AMPK activity in relevant tissues such as skeletal muscle and the plasma AMP/ATP ratio in the same hyperglycemic person’s blood sample may be important complementary information to insulin sensitivity and insulin resistance assays for more detailed diagnosis of DM. However, we acknowledge that our data were generated from plasma samples collected from a cross-sectional case control study with skewed age distribution between the groups, and thus the association of AMP elevation with DM is confounded by the older age in the DM patients.

Second, we observed that plasma levels of indole-3-propionic acid (IPA) and indole-3-lactic acid (ILA) are elevated in DM-N and DKD patients, both metabolized from tryptophan by gut flora. It was previously shown that, in a prospective study of at-risk Finnish individuals, higher levels of IPA were associated with lower risk of type 2 diabetes and serum CRP-based low-grade inflammation levels [34]. Similarly, ILA has been linked to amelioration of salt-sensitive hypertension [35]. In the context of cross-sectional case–control study, we can deduce that the elevated levels of IPA and ILA in the subjects with DM are likely the microbiome-mediated response to the treatments for improved glycemic control.

Last, we observed increased levels of a few lysophosphatidylethanolamine (LPE) and phosphatidylamine (PE) species and decreased levels of ether-linked phosphatidylethanolamine species as well as other lysophospholipids of different head groups in DM-N and DKD. It is likely that this change in balance in the phospholipid composition has to do with systemic dyslipidemia, and has little to do with localized perturbation of lipid efflux, uptake, and metabolism in kidney tissues in the case of DKD [36,37]. Although the methanol-based metabolite extraction still resulted in a decent coverage of eicosanoids, sterol lipids, phospholipids and some ceramides, a shortcoming of our current study is that we did not detect the majority of triglycerides and cholesteryl esters. Despite this drawback, the biochemical assay data in Table 1 suggest that only the total triglyceride level is elevated in DM and DKD, but LDL, HDL, and total cholesterol levels are more or less equivalent to the controls. We therefore speculate that even the composition of individual TG species would not have provided additional information delineating the difference between DM-N and DKD in the current study cohort. We leave this aspect of investigation to future research.

## 4. Materials and Methods

*Metabolite extraction.* In this process, 800 µL of ice cold methanol was added to 200 µL of each plasma sample. The mixtures were incubated at −20 °C for 60 min to precipitate proteins and centrifuge at 16,000 g at 4 °C for 10 min. The supernatants were divided into two aliquots and dried in a vacuum concentrator. Quality control (QC) samples were prepared by pooling equal volume of all plasma samples in this study to monitor the stability and repeatability during LC-MS analysis and the same protocol was used for metabolite extraction from QC samples.

*LC-MS/MS analysis*. Metabolite analysis was performed on an ACQUITY I-class UPLC system (Waters, Milford, MA, US) coupled to a TripleTOF 5600 fitted with a DuoSpray ion source (SCIEX, Foster, CA, US). Each sample was reconstituted in 30 µL of 95/5 water/methanol (*v/v*) and 5 µL was injected for each analysis in the positive mode and negative ionization mode, with both information-dependent acquisition (IDA) and sequential windows acquisition of all theoretical fragment ion mass spectra (SWATH-MS). Samples were injected in a randomized order and a QC sample was injected after every 10 samples. Mass calibration was automatically performed after every 20 injections by the automated calibration delivery system.

*Reverse phase liquid chromatography and positive ionization mode*. The column used for positive ionization mode was a Waters HSS T3 2.1 mm × 100 mm, 1.8 µm. Mobile phase A was 0.1% formic acid in 5% acetonitrile and mobile phase B was 0.1% formic acid in 95% acetonitrile. The gradient profile was 2% B from 0 to 1 min, 50% B at 8 min, 98% B from 13 min to 15 min and 2% B at 15.1 min to 20 min. The flow rate was set to 0.4 mL/min. The temperature of the column oven and auto-sampler was set to 40 °C and 4 °C, respectively. The source voltage was 5500 V.

Reverse phase liquid chromatography and negative ionization mode. The column used for analysis in the negative ionization mode was a Waters BEH C18 2.1 mm × 100 mm, 1.7 µm column. Mobile phase A was 5 mM ammonium bicarbonate in 5% acetonitrile and mobile phase B was 5 mM ammonium bicarbonate in 95% acetonitrile. The gradient profile was 2% B from 0 to 1 min, 50% B at 8 min, 98% B from 13 min to 15 min and 2% B at 15.1 min to 20 min. The flow rate was set to 0.4 ml/min. The temperature of the column oven and auto-sampler was set to 45 °C and 4 °C, respectively. The source voltage was 4500 V.

IDA (DDA) acquisition. Each IDA duty cycle contained one TOF MS survey scan (180 ms) followed by 20 MS/MS scans (40 ms). The mass ranges of TOF MS and MS/MS scans were 100 to 1000 and 30 to 1000 *m/z*, respectively. The following IDA parameters were applied: dynamic background subtraction, charge monitoring to exclude multiply charge ions and isotopes and dynamic exclusion of former target ions for s. Ramped collision energies of 20 to 40 V and −20 to −40 V were applied for positive and negative MS/MS scans, respectively.

SWATH-MS (DIA) acquisition. Each SWATH duty cycle contained one TOF MS survey scan (150 ms) followed by 36 SWATH scans (20 ms each). The fragment ion window for SWATH was from 100 to 1000 *m/z* in steps of 25 Da. The mass range of TOF MS and SWATH scans were 100 to 1000 and 30 to 1000 *m/z*, respectively. Ramped collision energies of 20 to 40 V and −20 to −40 V were applied for positive and negative MS/MS SWATH scans, respectively.

MetaboKit analysis [13]. IDA files were processed for spectral library construction using the NIST2014, HMDB [38], MassBank [39], LipidBlast (main and fork) [40] as reference libraries. For peak extract, we considered ion chromatograms spanning between 3 and 100 s. For compound identification, precursor ion *m/z* should match the theoretical monoisotopic mass within 15 ppm. For MS/MS-based scoring, the modified dot product score was required to be at least 0.5, and at least two fragment ions must match peaks in the reference spectra within 30 ppm. This process generates an MS1-based peak area table as well as a custom MS/MS library with matching records to the reference libraries.

For the SWATH-MS analysis, we trimmed the raw data by removing all peaks with an intensity value of 1000 or below. The length of ion chromatograms was limited to 3 to 100 s, as in the IDA analysis. Using the custom library created above [12], we rolled up the peak areas of the top six most intense fragment ions to derive a semi-quantitative value for individual compounds. Ion chromatograms of fragment ions were required to have at least 0.5 Pearson correlation with the ion chromatograms of the corresponding precursor ion, with the dot product score at the apex of the elution to be 0.5 and above. This time point was also required to be within 20 s from the RT marked for each record in the custom library.

Statistical analysis. Log-transformed (base 2) peak areas of precursor ions from the SWATH-MS analysis were used as semi-quantitative data for metabolites. Univariate differential expression analysis (t-test with multiple testing correction by *q*-value [41]), univaraite linear regression analysis, heatmap visualizations, and sparse Gaussian graphical model estimation (glasso) for deriving sparse inverse covariance matrix were performed using glasso package in R [16]. The estimated inverse covariance matrix was converted into a partial correlation matrix, and metabolite pairs (nodes) with non-zero partial correlations were connected by lines (edges) to form the partial correlation network.

Network-based multivariate classification analysis with iOmicsPASS. Using this network as the background, supervised classification analysis was performed by the iOmicsPASS software [17] to identify subnetwork signatures of DM-N and DKD groups from 87 and 59 samples, respectively, after dropping the outliers. In iOmicsPASS, the co-expression score of an edge for a given sample is derived from the z-scores of the two connected nodes. If the partial correlation between the nodes is positive, the edge score for that sample is calculated as the sum of the two z-scores. If the partial correlation is negative, the edge score is the difference between the two z-scores, whichever node was named first in the network input file. Hence, the interpretation of up and down between groups depends on the order of appearance of the nodes, and therefore it is important to reaffirm the abundance levels of individual nodes for proper interpretation in the latter case. The subnetworks for individual comparison groups (DM-N and DKD in Figure 3 and DKD in Figure 4) were then visualized using Cyotpscape software. In the networks, an edge was colored in red if the partial correlation between the two nodes was positive, and in blue if the partial correlation was negative. The thickness of edges was proportional to the magnitude of the group centroid values, i.e., the discriminative co-expression score reported by the supervised classifier in iOmicsPASS.

## Figures and Tables

**Figure 1 metabolites-11-00228-f001:**
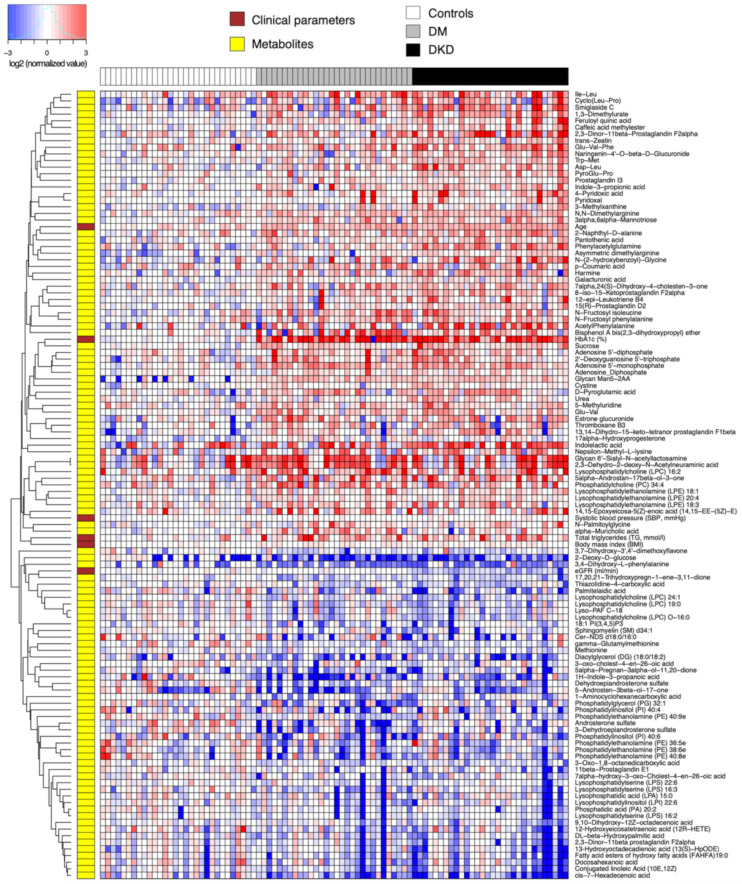
Heatmap of relative abundance values (log-transformed, base 2) of endogenous compounds differentially abundant between 60 DM patients and 30 controls. Data were normalized by the median of the 30 control samples in each metabolite.

**Figure 2 metabolites-11-00228-f002:**
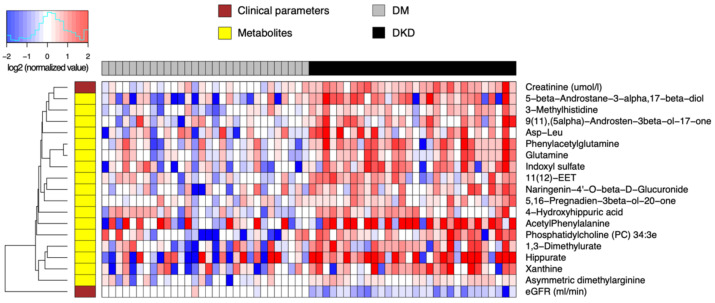
Heatmap of relative abundance values of endogenous compounds (log-transformed, base 2) differentially abundant between 30 DKD patients and 30 DM-N. Data were normalized by the median of the 30 DM-N samples in each metabolite.

**Figure 3 metabolites-11-00228-f003:**
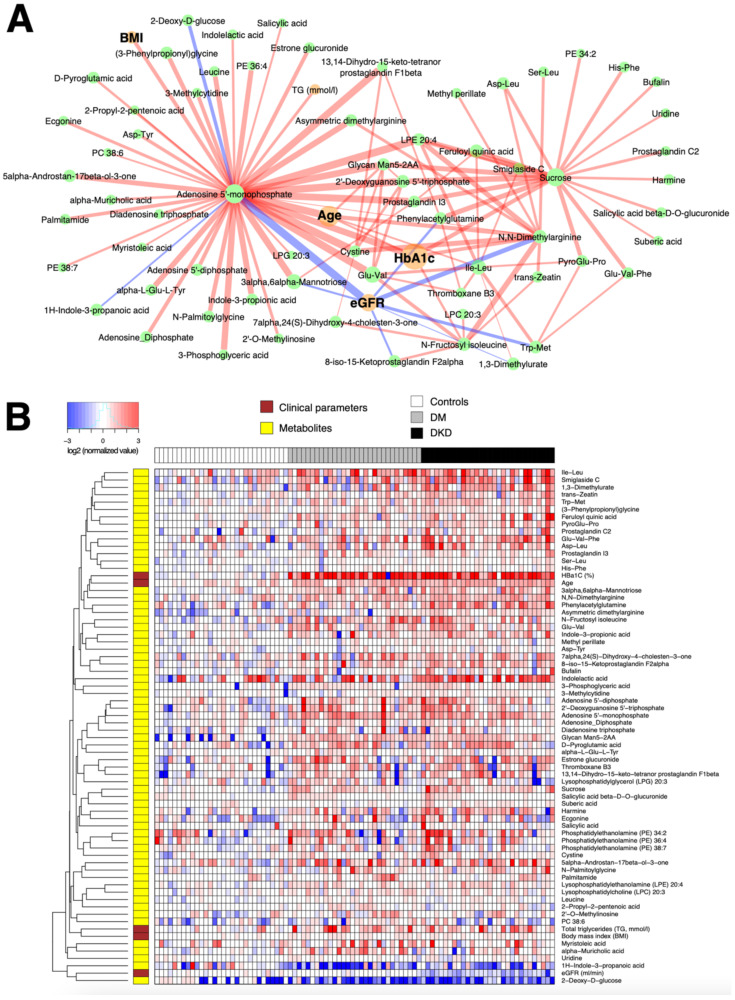
(**A**) Subnetwork signature of 60 DM patients compared to the 30 controls. Network edges are drawn in different colors depending on the sign of the partial correlations (positive in red, negative in blue); thicker edges represent stronger relationships. The size of the nodes corresponds to their respective −log10(q-value) from univariate differential abundance tests (two-sample t-test). (**B**) To demonstrate the abundance patterns more clearly, heatmap of relative abundance values of individual compounds was drawn (log-transformed, base 2) normalized by the median of the control group.

**Figure 4 metabolites-11-00228-f004:**
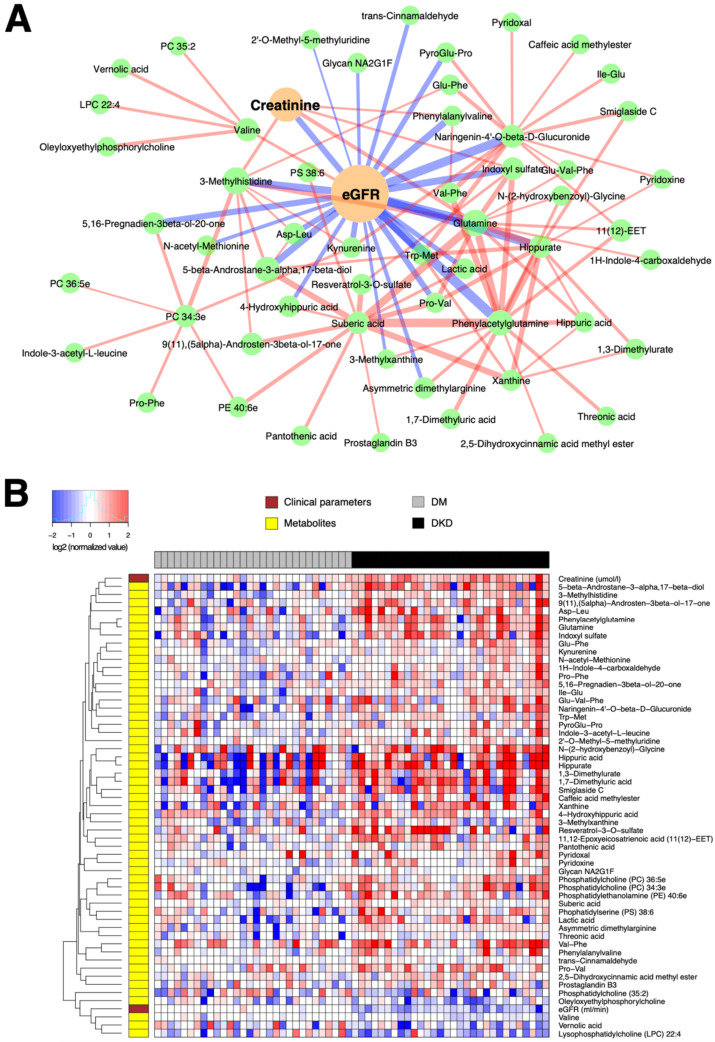
(**A**) Subnetwork signature distinguishing 30 DKD patients from 30 DM-N patients. Network edges are drawn in different colors depending on the sign of the partial correlations (positive in red, negative in blue); thicker edges represent stronger relationships. The size of the nodes corresponds to their respective −log10(q-value) from univariate differential abundance tests (two-sample *t*-test). (**B**) To demonstrate the abundance patterns more clearly, heatmap of relative abundance values of individual compounds was drawn (log-transformed, base 2), normalized by the median of DM-N, in which the plasma levels of a majority of compounds were elevated in DKD patients.

**Table 1 metabolites-11-00228-t001:** Patient characteristics and clinical biochemistry values in the controls, subjects with diabetes mellitus (DM) and normal renal function (DM-N), and subjects with diabetic nephropathy (DKD).

Variable	Unit/Level	Controls (N = 30)	DM-N (N = 30)	DKD (N = 30)	Statistical Significance
Age	Years	32.7 (10.2)	48.6 (10.6)	54.9 (7.0)	*p* < 0.001
Gender	Male Female	22 (73.3%) 8 (26.6%)	14 (46.7%) 16 (53.3%)	23 (76.7%) 7 (23.3%)	*p* = 0.028
Race	Chinese Indian Malay Other	30 (100.0%) 0 (0.0%) 0 (0.0%) 0 (0.0%)	15 (%) 4 (%) 9 (%) 2 (%)	17 (56.6%) 1 (3.3%) 9 (30.0%) 3 (10.0%)	*p* < 0.001
BMI	kg/m^2^	25.3 (3.4)	29.6 (3.2)	27.8 (4.0)	*p* = 0.014
SBP	mmHg	123.3 (13.9)	131.1 (13.6)	133.6 (14.9)	*p* < 0.001
HbA1c	%	5.3 (0.4)	8.6 (1.8)	8.5 (2.1)	*p* < 0.001
eGFR	mL/min/1.73m^2^	126.0 (23.4)	108.6 (12.3)	72.6 (16)	*p* < 0.001
sCR	μmol/L	75.3 (13.5)	59.7 (13.9)	101.7 (34.3)	*p* < 0.001
TC	mmol/L	4.7 (0.7)	4.9 (1.4)	4.7 (1.5)	*p* = 0.390
TG	mmol/L	1.1 (0.5)	1.8 (0.9)	1.9 (1.0)	*p* < 0.001
HDL	mmol/L	1.3 (0.3)	1.2 (0.3)	1.2 (0.3)	*p* = 0.101
LDL	mmol/L	2.9 (0.6)	2.6 (0.8)	2.7 (1.3)	*p* = 0.050

BMI—body mass index. SBP—systolic blood pressure. eGFR—estimated glomerular filtration rate. sCR—serum creatinine. TC—total cholesterol. TG—total triglycerides. HDL—high-density lipoprotein. LDL—low-density lipoprotein. For continuous variables, the numbers are averages and standard deviations (in parenthesis) per group. For categorical variables, the numbers are counts and percentages (in parenthesis) per group. For statistical significance values, the Kruskal–Wallis test was used for continuous variables and the chi-squared test was used for categorical variables.

## Data Availability

Interested users may contact the corresponding author for the access to raw mass spectrometry data (hyung_won_choi@nus.edu.sg).

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
