# Peer review of "Plasma Metabolome and Lipidome Associations with Type 2 Diabetes and Diabetic Nephropathy"

_metabolites, 2021, doi:10.3390/metabo11040228_

Round 1

Reviewer 1 Report

Tan et al. present a metabolomic analysis of diabetic kidney disease (DKD) in a study cohort of 30 controls, 30 persons with type 2 diabetes (T2D) and no DKD, and 30 persons with both T2D and DKD. The three study groups have major differences in their clinical characteristics (Table 1), including age difference in the range of years (between the conditions) or even decades (to the control group). Therefore, it is not possible to uncouple, how age (or other factors affecting metabolism), T2D, and DKD individually contribute to the metabolome. Regardless, the presentation of the results is descriptive and interesting, and the manuscript is well-written. I appreciate how the authors have found a good balance between exploring the data in interesting ways without inflating the interpretation of the findings beyond the limitations of the cohort.

Major Comments:

  1. The latter half of the Introduction Section appears to me more as Results and Discussion. The main task in Introduction should be to give a dive into what is known and what is not known about the metabolome in diabetic nephropathy.
  2. The Discussion Section is a solid package – from 5 years ago. The field of clinical metabolomics in diabetic kidney disease has expanded a great deal in the past 5 years and has moved on from small case-control studies to large cohorts. The results should be discussed in today’s context.
  3. The used network mining approach, iOmicsPASS, essentially is a multivariate classifier. I understand that, in this study, the method was used more as a descriptive tool rather than a predictor. Regardless, it should be made clear that the inferred model, representing relevant subnetwork(s), has not been validated with independent test data. – We do not know, how well the inferred subnetworks generalize to new data. This should be discussed as a limitation.
  4. Major differences in the characteristics of the study populations between the conditions and the control group should be discussed as a limitation. For finding clinically relevant new knowledge, the populations must be of comparable age. The DKD vs. no-DKD part is better balanced and thus suffering less from this limitation, which also could be indicated.

Minor Comments:

Abstract

  • It could help the reader, if “Supervised classification analysis over a confounding-free partial correlation network” was explained a bit more later in the text (Methods, primarily). In my impression, many readers are familiar with networks but perhaps not all with “predictive subnetworks”.

Introduction

  • Please clarify “data dependent and data independent acquisition mass spectrometry”. How is it dependent and independent at the same time? If you combine two approaches, I’d suggest making it more obvious here, and explain how.
  • “workflow was able to quantify 1,088 endogenous metabolites and lipids” – Is quantification an appropriate term here? Which unit of abundance are we talking about?

Results

  • L89: “control samples” => “participants in the control group”
  • Table 1: Write explanation of ”DM-N” and “DKD” in the list of abbreviations for improved readability -- even though they have been introduced earlier in the text.
  • “identified 1,233 unique compounds” – What level of identification certainty would you describe this as? Is term “identification” appropriate or would you use “annotation”?
  • “all quantification data are based on the peak areas of precursor ions” – The implications of this should be added to the Discussion Section.
  • Figure 1: The text are rather small but the heatmap gives a good overview of the data. Would it be possible to scale the figure to full page (minus the caption) to increase text size by some 50 %?`
  • Figures 1 & 2: There are some raw abbreviations that would be useful to write in human-readable form. Examples: “12R-HETE”, “11(12)-EET”, …
  • You have inferred many, many, non-zero edges in a network of 1088 metabolites from 90 samples. The regularization parameter in graph LASSO controls, how dense the network becomes. Which procedure did you use to objectively choosing the value for this parameter? Are the results stable and generalizable?
  • Figure 3 (A): q-value of which test?
  • Figure 4: eGFR is (by definition) decreased in DKD. According to the heatmap (B), most of the metabolites in the subnetwork are elevated in DKD. According to the network (A), all except one of the metabolites associated with eGFR are positively associated. That is, they go down when eGFR goes down. What is the reason for this apparent discrepancy between panels A and B? Is there something wrong with the fit of the network?

Discussion

  • (See major comment about the state of the field.)

Methods

  • L372: Incomplete sentence: “ions for s.”
  • L499: “univaraite"
  • L408: “Cyot- pscape”

The manuscript is well-presented and written with excellent language. I do not have any further remarks therein.

Thanks for an interesting and well-written manuscript!

Reviewer 2 Report

The peer-reviewed article is an interesting research, well-structured, well-chosen illustrative material. From the point of view of statistics, adequate methods were used, the conclusions are logical and well-grounded. I had a question about the formation of groups: how correct, in the opinion of the authors, is to use groups that differ in age and race? The age differences are quite significant: 32.7, 48.6 and 54.9 years. Are the differences identified, including those related to age? 
